# A Novel Strategy for Glioblastoma Treatment by Natural Bioactive Molecules Showed a Highly Effective Anti-Cancer Potential

**DOI:** 10.3390/nu16152389

**Published:** 2024-07-23

**Authors:** Alessandro Giammona, Mauro Commisso, Marcella Bonanomi, Sofia Remedia, Linda Avesani, Danilo Porro, Daniela Gaglio, Gloria Bertoli, Alessia Lo Dico

**Affiliations:** 1Institute of Molecular Bioimaging and Physiology (IBFM), National Research Council (CNR), Segrate, 20054 Milan, Italy; 2NBFC, National Biodiversity Future Center, 90133 Palermo, Italy; 3Department of Biotechnology, University of Verona, 15, Strada Le Grazie, 37134 Verona, Italy

**Keywords:** brain tumors, biodiversity, natural compounds, anticancer drugs, resistance

## Abstract

Glioblastoma (GBM) is a severe form of brain tumor that has a high fatality rate. It grows aggressively and most of the time results in resistance to traditional treatments like chemo- and radiotherapy and surgery**.** Biodiversity, beyond representing a big resource for human well-being, provides several natural compounds that have shown great potential as anticancer drugs. Many of them are being extensively researched and significantly slow GBM progression by reducing the proliferation rate, migration, and inflammation and also by modulating oxidative stress. Here, the use of some natural compounds, such as *Allium lusitanicum*, *Succisa pratensis*, and *Dianthus superbus*, was explored to tackle GBM; they showed their impact on cell number reduction, which was partially given by cell cycle quiescence. Furthermore, a reduced cell migration ability was reported, accomplished by morphological cytoskeleton changes, which even highlighted a mesenchymal–epithelial transition. Furthermore, metabolic studies showed an induced cell oxidative stress modulation and a massive metabolic rearrangement. Therefore, a new therapeutic option was suggested to overcome the limitations of conventional treatments and thereby improve patient outcomes.

## 1. Introduction

Biodiversity is the basis of individual well-being, as it provides essential resources (sources of food and water) while maintaining ecological balance [1]. The connection between the protection of biodiversity, sustainable development, and planetary health therefore becomes fundamental for the protection of human and planetary health. With this in mind, the National Biodiversity Future Center (NBFC) is developing some project activities aimed at protecting and enhancing biodiversity [2]. Among these, the study of bioactive molecules extracted from Italian flora promotes the protection of biodiversity and enhances the study of biomolecule functional activities, to enhance their use for human well-being.

Plants have been integral to human health and medicine since time immemorial, with a rich history of use in treating various diseases. Among these, cancer stands as a formidable challenge; yet, plants have emerged as a valuable resource in the quest for effective treatments. Hartwell (1982) extensively documented the utilization of over 3000 plant species in cancer treatments, marking a testament to their historical significance in medicinal practices.

A noteworthy aspect of plant-based medicine in cancer therapy lies in the discovery of potent anti-cancer agents derived from natural sources. Notably, over 60% of current anti-cancer drugs find their origins in nature, encompassing plants, marine organisms, and microorganisms [3].

This underscores the profound impact of natural compounds in modern pharmacotherapy.

Among the plethora of plant-derived anti-cancer agents, notable examples include Paclitaxel, Vincristine, Vinblastine, Irinotecan, Topotecan, and Etoposide [4,5].

These compounds have revolutionized cancer treatment, offering hope to countless patients worldwide.

However, the exploration of the plant kingdom’s therapeutic potential remains incomplete; while several efficacious anti-cancer drugs originate from plants, vast swathes of botanical diversity remain untapped.

In this framework, the NBFC aims to explore the under-exploited Italian flora through an ambitious program that aligns with bioprospection, metabolomics, and functional assays. Here, we present such an approach for discovering the potential of phytoextracts for anti-cancer activity with a first insight into the putative mechanism of action.

Glioblastoma multiforme (GBM) is a devastating disease that accounts for almost 80% of all malignant primary tumors of the brain. Despite significant advances in treatment, it remains a deadly condition with a poor prognosis. Patients diagnosed with GBM have a median survival of only 14.6 months, making it crucial to take preventative measures and seek early treatment. It is important to note that recurrence is almost inevitable, with an average progression-free survival of only 7 months from diagnosis. Therefore, timely and proactive measures should be taken to minimize the risk of recurrence and ensure better treatment outcomes, and of course, it has become crucial to raise awareness and encourage further research to find a cure [6,7,8].

Considering that most of the treatments nowadays are not curative, few randomized trials have addressed the question of the best treatment option, and an even higher number of novel and different approaches have been considered and developed to face GBM.

These include loco-regional treatments, such as surgery, radiotherapy (RT), immunotherapy, and chemotherapy (CT) such as nitrosoureas, antiangiogenic drugs (bevacizumab), tyrosine kinase inhibitors (TKI) and temozolomide (TMZ), which is indicated as the gold standard of the so-called STUPP protocol [9,10,11].

Nowadays, it has been explored to maximize efficacy or minimize the side effects of the treatment by combining several natural-based solutions with standard therapeutic approaches. Numerous in vitro and in vivo studies support the efficacy of plant extracts in some of the key pathological processes involved in the GBM aggressive features and its chemoresistance. For instance, plant extracts have been described as useful in modulating apoptosis and autophagy, increasing tumoral cell death, mitigating proliferation, cell cycle, metabolism, angiogenesis, and invasion, and counteracting all the biological mechanisms involved in tumor growth [12].

This article is based on research into territorial biodiversity and aims to identify the anti-tumor properties of certain biomolecules derived from plants. The observed properties could be valuable in developing new drugs for treating GBM, with the potential to induce sensitization to traditional treatment with temozolomide or act as adjuvant therapies for treating resistant glioblastoma.

## 2. Materials and Methods

### 2.1. Plant Species Preparation and LC-MS Analysis

*Allium lusitanicum* Lam., *Succisa pratensis* Moench, and *Dianthus superbus* L. subsp. *superbus* plants were purchased from a public organization, the “Centro Biodiversità Vegetale e Fuori Foresta” (Montecchio Precalcino, Vicenza, Italy), which aims to protect and preserve the germplasm of local plant species, and were grown in the greenhouse facility in the University of Verona.

Six plants of each species in vegetative growth were used to create three biological replicates, each composed of two plants for each replicate. Leaves were collected from *A. lusitanicum* on 21 July 2022 and from *S. pratensis* and *D. superbus* on 18 July 2022. The samples were processed as described in our previous work [13]. Briefly, the leaves were immediately frozen in liquid nitrogen, homogenized by using an A11 basic analytical mill (IKA-Werke, Staufen, Germany), and stored at −80 °C. The frozen powders (1 g) were extracted with 10 volumes (*w*/*v*) of methanol (LC-MS grade; Honeywell, Seelze, Germany), vortexed for 30 s, sonicated in an ultrasonic bath with ice for 10 min at 40 kHz (SOLTEC, Milano, Italy), and centrifuged for 10 min at 4 °C, at 14,000× *g*, and the supernatants were stored at −20 °C.

The methanol extracts were diluted 1:50 with LC-MS grade water (Honeywell), filtered through 0.22 mm Minisart filters (Sartorius-Stedim Biotech, Göttingen, Germany), and analyzed by following an untargeted metabolomics approach with a protocol described in a previous work [13]. The chromatographic method underwent minor modifications: it started with 1% B, held to 1% B for 1 min, then increased to 40% B at 10 min, to 70% B at 13.5 min, to 90% B at 15 min, and to 99% at 16.5 min. Subsequently, the method remained at 99% B for 3.5 min and was then decreased to 1% B at 20.1 min. The method remained in an isocratic form (1% B) and ended at 25 min. The UPLC-ESI/HRMS analysis was performed in both positive and negative ionization modes and 1 and 5 μL were injected, respectively. To better identify the metabolites, a FAST-DDA analysis was performed in both negative and positive ionization modes.

The putative metabolite identification was performed by considering the accurate mass, retention time, and fragmentation pattern of the main detected signals. These data were compared with those included in a library of authentic standard compounds and/or with data presented in the scientific literature and public databases (e.g., MassBank. HMDB, Pubchem, and MoNA).

In order to use the biomolecules for the subsequent cellular assays, they were then resuspended in pure DMSO at a final concentration of 0.5 mg/μL and used at a final concentration of 5 μg/mL.

### 2.2. Cell Lines and Reagents

The GBM cell lines U118, LN18, T98, and HEK-293T were purchased from ATCC. The U118, LN18, and HEK293T cells were maintained in Dulbecco’s modified Eagle’s medium (DMEM) (Euroclone, Milan, Italy) containing 10% FBS, 1% P/S, and 1% glutamine (Lonza, Basel, Switzerland) at 37 °C with 5% CO_2_ in a humidified environment. The T98 cells were cultured in RPMI (Euroclone, Milan, Italy) containing 10% fetal bovine serum (FBS) (Euroclone), 1% penicillin-streptomycin (P/S) solution (Thermo Fisher Scientific, Waltham, MA, USA), and 1% non-essential amino acids (Euroclone) at 37 °C in a 5% CO_2_ incubator.

### 2.3. Proliferation and Wound Healing Assays

For the proliferation assay, 50.000 GBM cells/well were seeded in a 24-well plate and cultured with biomolecules at 5 mg/mL. After 24 h, the cells were counted by automated cell counting (Diatech Lab Line, Jesi, Italy). A wound healing assay was performed using a 12-well plate, where a range number of GBM cells (from 50.000 to 80.000 cells/well) were seeded in their media to reach a 70–80% confluence after 24 h growth. After the cells reached complete confluence, the cell layer was scraped perpendicularly. The cell monolayer was then gently washed and the media was refreshed by adding biomolecules at a concentration of 5 mg/mL. After 24 h, the cells were imaged using a phase-contrast microscope. The area of the wound was quantified by Java’s ImageJ software (version 2.0.0-rc-43/1.50e) and the migration of cells toward the wounds was expressed as a percentage of wound closure: % of wound closure = [(A (t = 0 h) − A(t = 24 h))/A(t = 0 h)] × 100, where, A(t = 0 h) is the area of the wound measured immediately after scratching, and A(t = 24 h) is the area of wound measured 24 h after scratching. The GBM cells were plated in 6-well plates in the appropriate normal growth medium. For proliferation curves under nutrient deprivation conditions, the culture medium was replaced after 18 h with either normal growth medium, low-glutamine medium (0.5 mM Gln), or low-glucose medium (1 mM Glc). The cells were collected and counted after 24 h and 48 h.

### 2.4. Cell Cycle

Cell cycle analysis was performed on 1 × 10^6^ dissociated GBMs fixed in 70% ice-cold ethanol in a dropwise manner, dispensed while mixing gently in a vortex, and incubated on ice for 1 h.

The cells were incubated with 50 μg/mL propidium iodide (Sigma-Aldrich, St. Louis, MO, USA), 3.8 mmol/L sodium citrate (Sigma), and 10 μg/mL RNase (Sigma) for 1 h in the dark and at room temperature. The samples were acquired through a Calibur flow cytometer (BD Biosciences, East Rutherford, NJ, USA). All the data were analyzed using FlowJo^TM^ software (vesion 10)(TreeStar’s Flowjo Flow Cytometric Data Analysis Software, BD (Becton, Dickinson & Compan, East Rutherford, NJ, USA).

### 2.5. Immunofluorescence

A total of 20.000 GBM cells were seeded on round glass slides and cultured in adherence with their own media; after 24 h, the biomolecules were added to the culture media at the concentration of 5 mg/mL.

After 24 h, the cells were washed by PBS and fixed for 30 min at 37 °C with 4% paraformaldehyde (PFA). After that, the round glass slides were washed with PBS and permeabilized with PBS/FBS 5%/Triton 0.3% solution for 10 min at 4 °C. After washing with PBS, the cells were incubated with a blocking buffer composed of BSA 2% in PBS tween 0.1% for 30 min at room temperature. The cells were incubated overnight at 4 °C with the following primary antibodies: Tubulin (sc-5286 Santa Cruz, Dallas, TX, USA), and E-cadherin (SAB4503751, Sigma-Aldrich). Thereafter, the cells were labeled as 488 Alexa Fluor-conjugated secondary antibodies (Invitrogen, Waltham, MA, USA) for 30 min at room temperature in the dark. The nuclei were blue counterstained with Hoechst33342 (H3570, Invitrogen). Next, the cells were washed with PBS, and the slides were closed with glass coverslips using a clear mount solution (Invitrogen). Images were acquired at oil 100× magnification with a Nikon Eclipse 80i fluorescence microscope (Nikon Corp., Tokyo, Japan). Cell morphological analysis was performed by using the ImageJ software (NIH, New York, NY, USA).

### 2.6. Quantitative Real-Time PCR

Total RNA was isolated using the TriPure isolation reagent (Roche, Basel, Switzerland) and subjected to DNase I treatment (Roche Diagnostics Indianapolis, Indiana, United States). Reverse transcription was performed using a high-capacity cDNA reverse transcription kit (Applied Biosystems, Carlsbad, CA, USA). qPCR amplification was carried out at 60 °C using FastStart SYBR Green Master (Roche Diagnostics, Indianapolis, IN, USA), in an AriaMx real time PCR system (Agilent Technologies, Santa Clara, CA, USA). Target mRNA content changes in the β-actin housekeeping gene were determined using the Delta Delta Ct method and represented as FOI (fold induction) compared to the control levels (vehicle). The primer sequences used for qPCR are listed in Appendix A.

### 2.7. Biochemical Assays

An ELISA-based kit (TransAM Kit, Vinci-Biochem, Vinci, Italy) was used to detect and quantify HIF-1α transcriptional factor activity according to the manufacturer’s instructions. The data are expressed as the amount of HIF-1α protein in the nuclear and cytoplasmatic extracts (optical density 450 nm). The cytotoxicity of the treatments was tested utilizing the Cell ToxGreen Cytotoxicity Assay kit and the Cell Titer-Glo^®^ Luminescent Cell Viability Assay (all Promega, Milan, Italy). The ROS content after all the treatments was tested by using the ROS-Glo H_2_O_2_ Assay kit (Promega, Milan, Italy). All the assays performed by using commercially available kits were carried out according to the manufacturer’s instructions.

### 2.8. Metabolite Extraction from Cell Culture and LC-MS Metabolic Profiling

The GBM cell lines (U118, LN18, and T98) were plated in 6-well plates with the above-described culture medium. After 24 h, it was replaced with the complete fresh medium in the presence or the absence of biomolecules (*A. lusitanicum, S. pratensis*, and *D. superbus*) at a 5 mg/mL concentration, then incubated for 24 h. Metabolite extraction for LC-MS analysis was performed as described previously [14]. Briefly, the cells were rinsed with NaCl 0.9% and then quenched with an ice-cold solution of 70:30 acetonitrile/water. The plates were placed at −80 °C for 10 min and then collected by scraping and were sonicated twice for 5 s for 5 pulses at 70% power. The samples were centrifuged at 12,000× *g* for 10 min, and the supernatant aqueous phases were collected in a glass insert and dried in a centrifugal vacuum concentrator (Concentrator plus/Vacufuge plus, Eppendorf, Hamburg, Germany) at 30 °C for about 2.5 h. The samples were then resuspended with 150 μL of H_2_O before the analyses. LC separation was performed using an Agilent 1290 Infinity UHPLC system and an InfintyLab Poroshell 120 PFP column (2.1 × 100 mm, 2.7 μm; Agilent Technologies, Santa Clara, CA, USA). Mobile phase A was water with 0.1% formic acid. Mobile phase B was acetonitrile with 0.1% formic acid. The injection volume was 10 μL, and the LC gradient conditions were 0 min: 100% A; 2 min: 100% A; 4 min: 99% A; 10 min: 98% A; 11 min: 70% A; 15 min: 70% A; and 16 min: 100% A, with 2 min of post-run. The flow rate was 0.2 mL/min, and the column temperature was 35 °C. MS detection was performed using an Agilent 6550 iFunnel Q-TOF mass spectrometer with a Dual JetStream source operating in negative ionization mode (Agilent Technologies, Santa Clara, CA, USA). The MS parameters were gas temp: 285 °C; gas flow: 14 L/min; nebulizer pressure: 45 psig; sheath gas temp: 330 °C; sheath gas flow: 12 L/min; VCap: 3700 V; Fragmentor: 175 V; Skimmer: 65 V; and Octopole RF: 750 V. Active reference mass correction was conducted through a second nebulizer using masses with *m*/*z*: 112.9855 and 1033.9881. Data were acquired from m/z 60 to 1050. Data analysis and isotopic natural abundance correction were performed with the MassHunter ProFinder and MassHunter VistaFlux software (version 10.0) (Agilent Technologies, Santa Clara, CA, USA), as described in [15].

### 2.9. Statistical Data Analysis

The in vitro experiments were repeated at least three times and led to reproducible results.

The data are presented as the mean values ± SD of the independent experiments and were statistically analyzed using a *t*-test or one- or two-way analysis of variance, followed by Dunnett’s or Bonferroni’s multiple comparisons, and Prism 4 software (GraphPad Software Inc., San Diego, CA, USA). The metabolomics data were analyzed using Mass Profiler Professional 15.1 software (Agilent Technologies). The raw data underwent an initial transformation in log2 scale. Following this, normalization was carried out using the protein content as an external scalar. Subsequently, the data were scaled using the Pareto scaling method. The data were then filtered, retaining in the analysis the entities that were at least present in 100% of the samples in one condition. Statistical analysis was performed by applying one-way ANOVA analysis with a *p*-value cut-off of 0.05. Multiple testing corrections when computing *p*-values were performed using the Benjamini and Hochberg false discovery rate. Data visualization of the significant entities was performed using a hierarchical clustering algorithm, which allows the visualization of normalized intensity values and the clustering of both entities and conditions with similar metabolic fingerprints.

## 3. Results

### 3.1. Natural Extract Characterization

Leaf methanol extracts of *Allium lusitanicum* Lam., *Succisa pratensis* Moench, and *Dianthus supersbus* L. subsp. Superbus were analyzed by Ultra-Performance Liquid Chromatography–High-Resolution Mass Spectrometry (UPLC-HRMS) following an untargeted metabolomics approach. The chromatograms of the samples ionized in a negative ion mode are shown in Figure 1.

The chromatograms of the three species may be split into two different sections, depending on the polarity range of the eluting molecules: mild-polar metabolites, including phenylpropanoids and secoiridoids, elute in the first section (blue rectangles in Figure 1), whereas low-polar metabolites, such as saponins, elute later (red rectangles in Figure 1).

In our precedent study, we reported that *A. lusitanicum* mainly accumulates 25 different metabolites, mostly belonging to the class of saponins [16]. The identities of these metabolites are reported in Appendix A, together with those belonging to *S. pratensis* and *D. superbus* subsp. Superbus.

The phytochemical profile of *S. pratensis* revealed the presence of 21 abundant metabolites. In detail, the phenylpropanoids eluted in the middle part of the chromatogram and included caffeic acid derivatives, the *C*-glycosylated flavones apigenin and luteolin, and secoiridoids glucosides, such as oleoside, swertiamarin, gentiopicroside, and sweroside. In the second part of the chromatogram, *S. pratensis* accumulated a specific saponin that was putatively annotated as Akebia saponin D based on the comparison of its fragmentation pattern with that published in a previous work [17].

The analysis of the *D. superbus* subsp. Superbus extract reveals the presence of 34 abundant metabolites. The mild-polar section was mainly characterized by two chromatographic peaks corresponding to an unidentified metabolite and two co-eluting *C*-glycosylated flavones (probably luteolins), respectively. In the low-polar section, *D. superbus* leaves accumulated saponins with gypsogenic and polygalacic acids as aglycones [18].

### 3.2. A. lusitanicum, S. pratensis, and D. superbus Treatment Decreased GBM Cell Number, Induced a Cell Cycle Quiescence, and Reduced Cell Migration

To identify the best biomolecule candidates for our aim, we first assessed extract cytotoxicity in a panel of three different GBM TMZ-resistant cell lines, T98, U118, and LN18. The biochemical assay showed no significant cytotoxic effect for any of the extracts screened (Figure 2A). After testing for no cell toxicity of the plant-derived extracts, we investigated their effect on cell proliferation by a trypan blue exclusion test cell count. It was found that three phyto-complexes from *A. lusitanicum*, *S. pratensis*, and *D. superbus* caused a significant proliferation rate reduction in GBM cell number when treated for 24 h with a final concentration of 5 mg/mL but did not show the same effect on HEK 293T cells, which were used as control cell lines (Figure 2B and Appendix A). This cell number reduction was also confirmed through a further cell viability assay based on fluorescent ATP measurement (Appendix A).

Thus, a cell cycle assay was conducted to confirm the antiproliferative effect of plant extracts on GBM cell lines. The effect was confirmed by observing an increase in the percentage of cells in the G0/G1 phase when the U118 cells were treated with *A. lusitanicum*, *S. pratensis*, and *D. superbus*. The same increase was also observed once the LN18 cells were treated with *A. lusitanicum* and *D. superbus*. However, we did not verify the G0/G1 arrest in the T98 cells treated with any of the three extracts. Instead, a strong cell percentage increase in the S cell phase was found when the T98 cell lines were treated with *S. pratensis*, and a slight increase was found when the same cells were cultured with *A. lusitanicum* and *D. superbus.* The same strong increasing effect was shown in the LN18 cells treated with only *S. pratensis* (Figure 2C).

Additionally, by wound-healing assay, further evidence of the anticancer properties belonging to *A. lusitanicum*, *S. pratensis*, and *D. superbus* was provided. It was observed that all the extracts significantly reduced the migration of all three GBM cell lines (Figure 2D and Appendix A).

Moreover, the key genes involved in apoptosis were evaluated by qPCR analysis, and an antitumoral effect by the plant extract treatment was confirmed. The apoptotic gene expression increased, whereas the antiapoptotic genes decreased, as shown by the BAX/BCL2 and BAD/BCL2 ratio (Appendix A).

### 3.3. Plant Extract Treatment Promoted Cytoskeleton Changes and Succisa pratensis Promoted a Mesenchymal–Epithelial Transition (MET)

To investigate the impact of selected plant extracts on GBM cell lines, the cells’ morphology was analyzed by performing immunofluorescence staining on cytoskeleton filaments and E-cadherin expression. Our fluorescence microscopy image analysis revealed that all the cells treated with plant extracts displayed a cellular morphology change, as shown by the reduced number of cytoskeleton filaments stained in green (Figure 3A). The highest effect was observed in the U118 GBM cell line treated with all three biomolecules, then in the T98 cells; only the treatment with *S. pratensis* and *D. superbus* showed an effect on the cytoskeleton filaments.

In addition to the morphologic change, once the GBM cells were treated for 24 h with plant extracts, it was possible to observe by immunofluorescence staining an increase in E-cadherin expression in those cells, which indicated an induction of MET (Figure 3B). In particular, the highest effect was in the U118 and LN18 cell lines treated with the *S. pratensis* extract. This result was confirmed by qPCR assay on E-cadherin and SNAIL gene expression. We observed an increase in E-cadherin gene expression and a SNAIL decrease due to *S. pratensis* treatment in the same GBM cell lines. (Appendix A).

### 3.4. A. lusitanicum, S. pratensis, and D. superbus Treatments Induced Cell Oxidative Stress Modulation

An assessment of the oxidative stress levels of the GBM cells cultured for 24 h with the three selected plant extracts (*A. lusitanicum*, *S. pratensis*, and *D. superbus*) was conducted, measuring the ROS levels. The results suggested a global increase in ROS levels in all the treated cell lines (Figure 4A).

Moreover, wide molecular analysis conducted by qPCR assay supported these results, showing that genes involved in detoxification, such as NRF2, SOD, GSH, and catalase, were impaired by the treatment, and at the same time, the NRF2 inhibitor KEAP’s gene expression increased (Figure 4B).

To deeply investigate the biomolecule effects on cancer cells’ antioxidant metabolism, we assessed using LC-MS analysis. The assay highlighted an imbalance of antioxidant machinery in all the GBM cells after treatment through the reduction in the GSH/GSSG ratio (Figure 4C).

### 3.5. Plant Extract Treatment Downregulated HIF-1α and NF-kB Expression and Induced the Reduction in Their Downstream Target Genes

Oxidative stress is known to stimulate certain regulatory pathways that are involved in cell proliferation and aggressiveness, such as HIF-1α and NFkB. A qPCR analysis to investigate the expression of both these genes was performed, and it was found that the gene levels were decreased in most of the GBM samples treated with biomolecules, as shown in Figure 4D,F.

In addition, the levels of HIF-1α at the proteomic level were also assessed using an ELISA test. As confirmation of the qPCR results, we observed a downregulation of nuclear HIF-1α localization and an increase in the cytoplasmatic localization as an inactive form (Figure 4E).

qPCR was used to investigate the expression levels of downstream HIF-1a and NFkB target genes, defined as master regulators of inflammasome machinery. Among the genes analyzed, IFNγ, IL18, and IL1β showed a significant decrease in expression levels in all the GBM cell lines due to the treatment with biomolecules (Figure 4G and Appendix A).

### 3.6. A. lusitanicum, S. pratensis, and D. superbus Induced a Metabolic Rearrangement

The observed increase in oxidative stress in the GBM cell lines, under natural extract treatments, prompted us to perform untargeted metabolomics mass spectrometry analysis to better investigate the metabolic impact of the biomolecules.

First, the preliminary metabolic characterization of the nutrient dependency of these cell lines was assessed by culturing these cells in low-glucose or low-glutamine conditions. The growth curves showed a glucose addiction in all the cell lines and a slight glutamine addiction in the T98 cells (Appendix A).

Hierarchical clustering of the metabolomics analysis of the control cells revealed a more similar metabolic signature between LN18 and U118 compared to T98 (Appendix A).

In addition, it was noticed that the T98 cell line, compared to LN18 and U118, showed a significant downregulation of metabolites involved in purine and amino acid metabolism. In particular, it was interesting to note that the levels of several metabolites related to the Warburg effect, such as citric acid, lactic acid, oxoglutaric acid, succinic acid, glutamine, fructose 1,6-bisphosphate, glucose 6-phosphate, GDP, and GTP, were significantly lower (Appendix A). Subsequently, metabolomics analyses of the effects of the plant extracts on each cell line were performed. The hierarchical clustering generated for U118 and LN18 showed identical clustering under different conditions.

Indeed, in both analyses, similar metabolic signatures between the control conditions and the allium-treated samples were found, highlighting a weaker metabolic effect of *A. lusitanicum* compared to *D. superbus* and *S. pratensis*, the latter having the highest degree of metabolic variations. Conversely, in the T98 cell line the first main cluster branch differentiated the control condition from the treated ones. In the sub-cluster of treated cells, *A. lusitanicum* and *D. superbus* exhibited a more similar metabolic signature when compared to *S. pratensis* (Figure 5).

All the cell lines treated with *A. lusitanicum* displayed significantly increased levels of serine, glycine, and the folate cycle intermediate (i.e., 5,10-Methenyl-THF and 5,10-Methylene-THF), which are all part of one-carbon metabolism; of the antioxidant molecules hypotaurine and taurine; and of oxidized glutathione, indicating a greater and coordinated response to oxidative stress. Several other amino acids (i.e., glutamine, threonine, and valine) and AMP had higher levels in the allium-treated cells than in the control.

In the LN18 cell line treated with the Allium extract, increased levels of metabolites involved in purine metabolism (cAMP, inosinic acid, hypoxanthine, GDP, ADP) and the TCA cycle (cis-aconitic acid; citric acid; fumaric acid; malic acid) were also observed.

It is worth noting that both pathways were upregulated, even in the T98 cell line. Additionally, a higher enrichment of one-carbon metabolism was found in the T98 and U118 treated cells, showing even higher levels of methionine cycle intermediates (i.e., methionine and S-adenosylhomocysteine).

Different metabolic behaviors in each cell line resulted after *D. superbus* treatment. The only common characteristic among the treated cell lines was increased levels of GSSG. In LN18 and T98, a downregulation of the first branch of glycolysis and pentose phosphate pathway was observed, along with higher levels of lactate compared to the control. Additionally, increased levels of the TCA cycle and one-carbon metabolism intermediates in the T98 cells were found. Conversely, U118 showed significantly decreased levels of metabolites involved in the TCA cycle and the glycine and serine metabolism, while the folate cycle cofactors—5,10-Methenyl-THF and 5,10-Methylene-THF—and the amino acids, aspartate, asparagine, and glutamine, resulted in upregulation (Figure 5).

The *S. pratensis* treatment exhibited similar effects across the three GBM cell lines. Increased levels of nutrients such as glucose and glutamine were noted, as well as antioxidant molecules like hypotaurine and carnosine. Additionally, the increased levels of oxidized glutathione, along with the nucleotides AMP and GMP and the folate intermediates, confirmed an activation of a response to oxidative stress. In detail, metabolic profiling showed several upregulated pathways in LN18, such as the TCA cycle, PPP, purine metabolism, and one-carbon metabolism. The latter resulted in upregulation even in the T98 Succisa-treated cells. Conversely, in U118 decreased levels of serine and glycine and of the TCA cycle intermediates were found, along with pyrimidine pathway metabolites (i.e., dCMP, UDP, UDP-glucose, UDP-glucuronate, UMP). Instead, purine metabolism was upregulated.

Taken together, these data show that the tested plant-derived compounds may induce an imbalance between antioxidant defense and pro-oxidant load, causing an increased threshold of reactive oxygen species (ROS), leading to oxidative stress as an effective strategy for cancer cell damage.

## 4. Discussion and Conclusions

This work is based on the biodiversity study to investigate and discover whether certain phytoextracts from plants and natural sources could possess promising antitumoral properties and consequently should be used to develop new efficacy drugs to efficiently impair cancer behavior. By a wide sampling and characterization of the following relative extracts, based on metabolic investigation, we selected a significant number of promising candidates to screen by biological assay and by the positive effects.

The first assays, performed in different GBM cell lines, were resistant to the primary chemotherapy treatments, such as temozolomide (TMZ), and allowed us to focus on three excellent candidates: *Allium lusitanicum, Succisa pratensis*, and *Dianthus superbus.*

The leaf extracts from these plants showed interesting metabolic profiles that highlighted the presence of saponins in all the selected plant extracts. Saponins are a specific class of secondary metabolites comprising bioactive glycosides with a steroidal or triterpenoid aglycone backbone. These secondary metabolites play a significant function in plant defense against pathogen attack or herbivore predation because of their bitter, astringent flavor and toxicity. Saponins are pharmaceutically used for their anti-thrombic, anti-inflammatory, anti-diabetic, anti-hypertensive, and anti-cancer activities [19]. The saponins detected in the selected plant samples comprise both steroidal and triterpenoidal types. In particular, *A. lusitanicum* presents steroidal saponins, while *S. pratensis* and *D. superbus* present triterpenoidal saponins. Steroidal saponins have been shown to inhibit various fundamental processes in cancer cells, such as initiation, growth, and metastasis, by targeting a number of molecules and signaling pathways. Indeed, triterpenoidal saponins have been shown to possess potential activity against various forms of cancers, including drug-resistant and MDR cancers [19] by triggering different mechanisms of action. Interestingly, while *A. lusitanicum* primarily exhibits steroidal saponins, *S. pratensis* and *D. superbus* predominantly contain triterpenoid saponins along with *C*-glycosylated flavones. This suggests that triterpenoid saponins may require flavones to display anti-cancer activity comparable to that of steroidal saponins, potentially through additive or synergistic effects. As a matter of fact, luteolin and apigenin, the two flavones present in *S. pratensis* and *D. superbus* have been described as anti-cancer agents against several types of cancer by suppressing tumor development and progression [20,21]. Previous studies have shown that luteolin can induce intrinsic apoptosis by disrupting mitochondrial membrane integrity, which leads to the release of cytochrome c and the activation of caspase-9. Additionally, luteolin can induce extrinsic apoptosis by promoting the expression of death receptors and the activation of caspase-8 [22]. Similarly, apigenin has been found to induce both extrinsic and intrinsic apoptosis. Furthermore, both luteolin and apigenin can trigger the unfolded protein response (UPR) by inducing the expression of genes associated with endoplasmic reticulum (ER) stress and increasing the expression of microtubule-associated protein light chain-3 (LC3) II, leading to the accumulation of autophagosomes [23].

Natural extracts suggest a cytostatic effect with the acquisition of a less aggressive phenotype. As shown in Figure 2A–C, all the selected plant extracts impaired the GBM proliferation rate, promoted cell cycle quiescence, and reduced cell migration. Moreover, as suggested by our data, the three plant extracts’ effects show different ways to promote their biological consequence. In particular, *S. pratensis* showed the highest effect on proliferation rate reduction compared with *A. lusitanicum* and *D. superbus*, but *A. lusitanicum* showed the best outcome in terms of migration ability reduction. Moreover, the cell cycle was differently impaired by the three molecules and even between the three GBM cells treated. *S. pratensis* strongly promoted the cell cycle arrest in the S phase in T98 and LN18; in contrast, the *A. lusitanicum* and *D. superbus* induced a G0/G1 block in LN18 and U118.

The groundbreaking research has provided compelling evidence that the selected plant extracts possess exceptional antitumoral potential to cause cytoskeleton disorganization in GBM cells. Figure 3A reports that treatment with biomolecules can significantly reduce microtubules. The immunofluorescence assays confirmed that the best anti-tumor effect was obtained by the *S. pratensis* treatment, especially in two out of the three cell lines, U118 and T98.

In addition, as reported in Figure 3B, we revealed that some biomolecules were even able to induce a mesenchymal–epithelial transition (MET), as shown by the increased positivity to E cadherin staining.

Therefore, as seen in the other reported assay results, the biological effect of three biomolecule treatments is not so consistent, even within the three TMZ-resistant GBM cell lines; therefore, it is necessary to investigate in detail the biomolecule peculiarities.

It has been hypothesized that this discrepancy could be attributed to the different metabolites found in the plant extract.

The treatment modulates oxidative stress by reducing key mediators of tumor progression, such as HIF-1, NF-kB, and inflammasome.

Hence, the investigation of the biological effect of the plant extract treatments focused on the triggered molecular mechanisms in the GBM cell lines, revealing a significant role of HIF-1α and NF-kB, particularly in reducing downstream inflammation target genes.

The reduction in inflammatory mediators supports the anti-tumoral effect of natural molecules. In fact, in tumors, and more specifically in glioma, the increase in the inflammatory process is associated with tumor progression and the malignancy of the pathology.

Additionally, a direct effect of the treatments on promoting oxidative stress through increased ROS levels was demonstrated. Figure 4 shows the decrease in expression of HIF, NFkB, and inflammasome genes, suggesting that *A. lusitanicum*, *S. pratensis*, and *D. superbus* have the potential to counteract the detrimental effects of cancer presence.

This work also aimed to investigate the metabolic effect of biomolecules on GBM cells. All the tested extracts caused a decrease in the ratio of GSH to GSSG (Figure 4C) in the three cell lines. This decrease was mainly due to an increase in the levels of GSSG in all the treated cells, suggesting that the redox state of the cells was altered as a result of the treatments. In most of the treated cells, one-carbon metabolism resulted in upregulation. This pathway promotes the increase in the production of NADPH, which provides reducing equivalents for the detoxification of reactive oxygen species (ROS), catalyzing the reduction of GSSG to GSH and powering cells [24]. This upregulation, along with an increase in other antioxidant molecules (such as taurine, hypotaurine, and carnosine), is the cell’s attempt to restore the balance of the redox system after treatment with the extracts.

### 4.1. Critical Issue and Final Remarks

In addition to some common traits, many differences in metabolic responses are evident between the different lines, even when using the same extract. This may be justified by the metabolic heterogeneity of the cell lines themselves, leading to different responses to the same treatment.

In the end, considering the high aggressiveness of GBM and its treatment failure due to chemotherapy and radiation resistance, scientists were forced to develop innovative therapeutic approaches, such as immunotherapy, targeted therapy, gene therapy, and metabolic targeting therapy [25,26,27,28].

Therefore, collecting all the obtained results from our work, we would like to consider the plant extract treatment a reliable approach to overcoming GBM resistance. Furthermore, our approach exploited natural extracts obtained just by a wide biodiversity study and also observed an antiproliferative effect and a metabolism change; in addition, these natural extracts could easily overcome the issue of the blood–brain barrier and enter into the tumor site [29].

### 4.2. Future Perspectives

Nevertheless, we are aware that our study represents only a first step for this type of approach and that these data must be strengthened by further bigger and deeper tests, such as animal testing that better simulates the physiological dynamics of a human tumor.

In addition, by chromatogram analysis our plant extracts showed the presence of different metabolites, and as a consequence, our next step will be to individually isolate each one and then test them one by one in GBM cells to properly link the observed effect described by our paper.

### 4.3. Highlighting Points

The treatment with bioactive molecules suggests:A reduction in the number of cells with non-significant cytotoxicity: potential cytostatic effect (as also suggested by the cell cycle);An increase in oxidative stress with consequent reduction in active HIF-1 form;A decrease in NF-kB and inflammasome.

## Figures and Tables

**Figure 1 nutrients-16-02389-f001:**
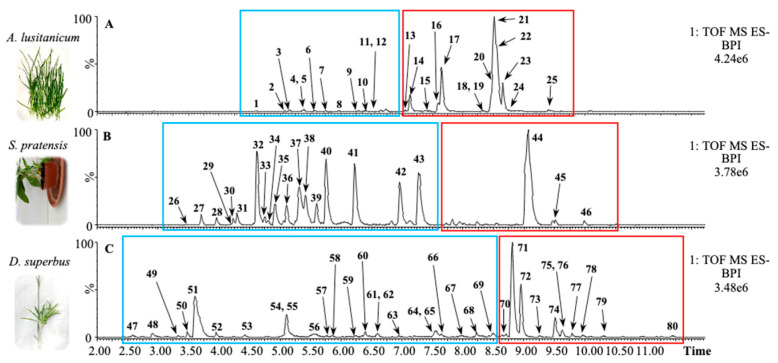
Base peak chromatograms (BPC) of Allium lusitanicum (**A**), Succisa pratensis (**B**), and Dianthus superbus subsp. Superbus (**C**) leaf methanol extracts. The chromatograms showed the main metabolites ionized in negative. The blue rectangles include the sections in which mild-polar metabolites, such as phenylpropanoids and secoiridoids, elute, whereas the red rectangles show the low-polar metabolites, such as saponins. Numbers indicate the putatively identified metabolites reported in Appendix A.

**Figure 2 nutrients-16-02389-f002:**
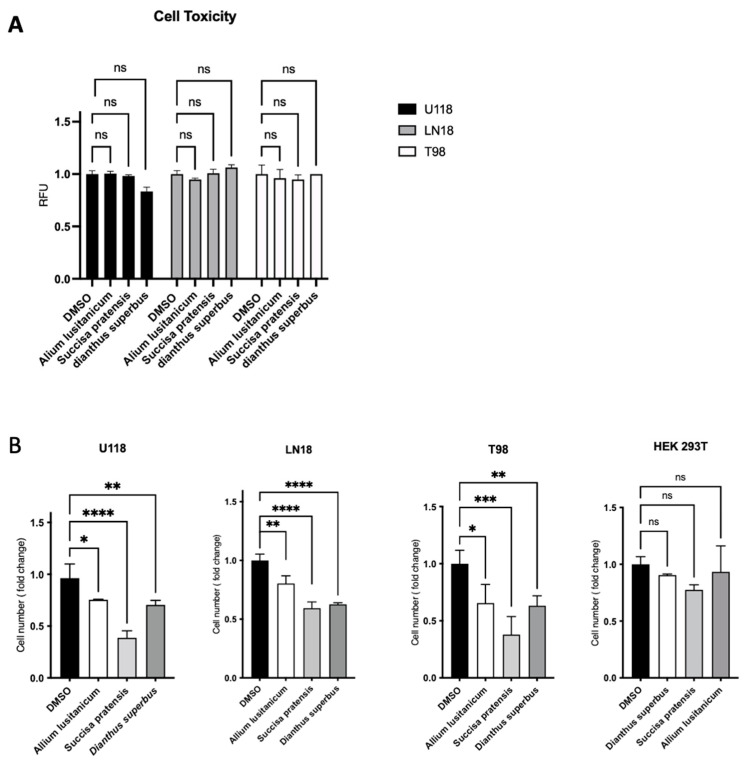
The biomolecules promoted an antiproliferative and anti-migratory effect on GBM cells. (**A**) Cell toxicity assay on resistant glioblastoma cell lines. (**B**) Proliferation assay by manual cell viability counting assay after 24 h on GBM- and HEK-293T-treated cells. Data represent mean ± SD of three independent experiments. * *p* < 0.05; ** *p* < 0.01; *** *p* < 0.001; **** *p* < 0.0001; (one-way ANOVA) (**C**) Propidium iodide cell cycle assay performed by flow cytometry analysis on all the GBM cells. (**D**) Representative graph of wound healing assay performed on the three GBM cells treated for 24 h. Data represent at least three independent experiments. * *p* < 0.05; *** *p* < 0.001; (one-way ANOVA).

**Figure 3 nutrients-16-02389-f003:**
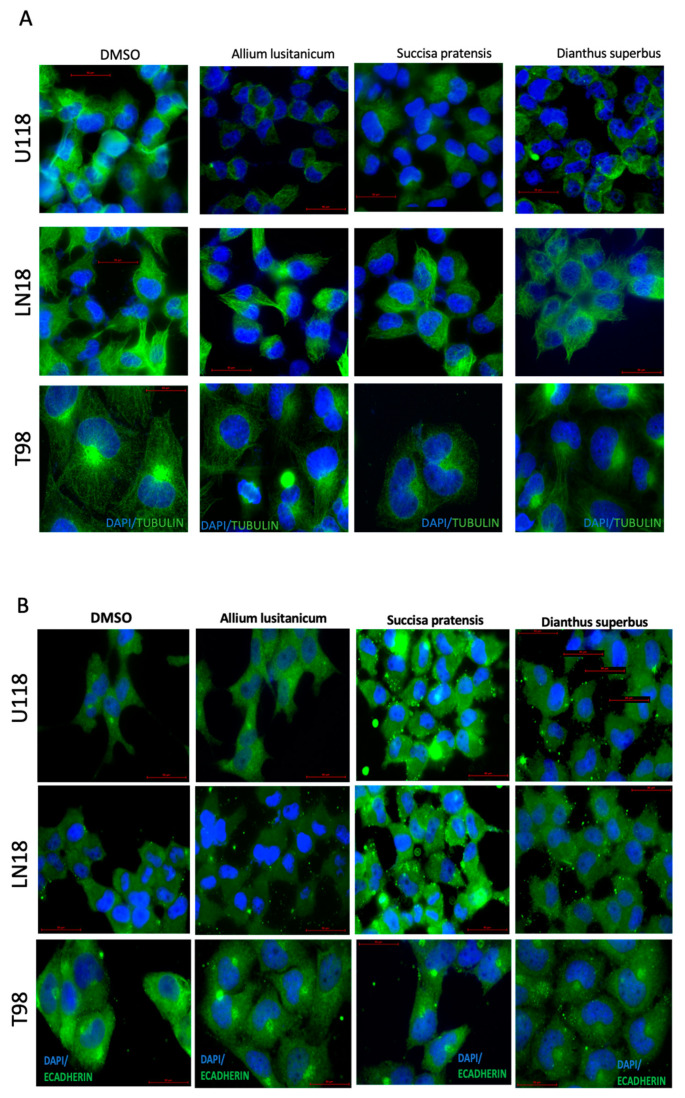
Biomolecules induced cytoskeleton rearrangement and promoted a mesenchymal–epithelial transition. Representative immunofluorescence images of GBM cells cultured for 24 h in their media supplemented with *Allium lusitanicum* Lam., *Succisa pratensis* Moench, and *Dianthus superbus* L. subsp. *Superbus*. The cells were stained with Tubulin (**A**) and E-cadherin (**B**) antibodies. Nuclei were blue counterstained with Hoechst33342.

**Figure 4 nutrients-16-02389-f004:**
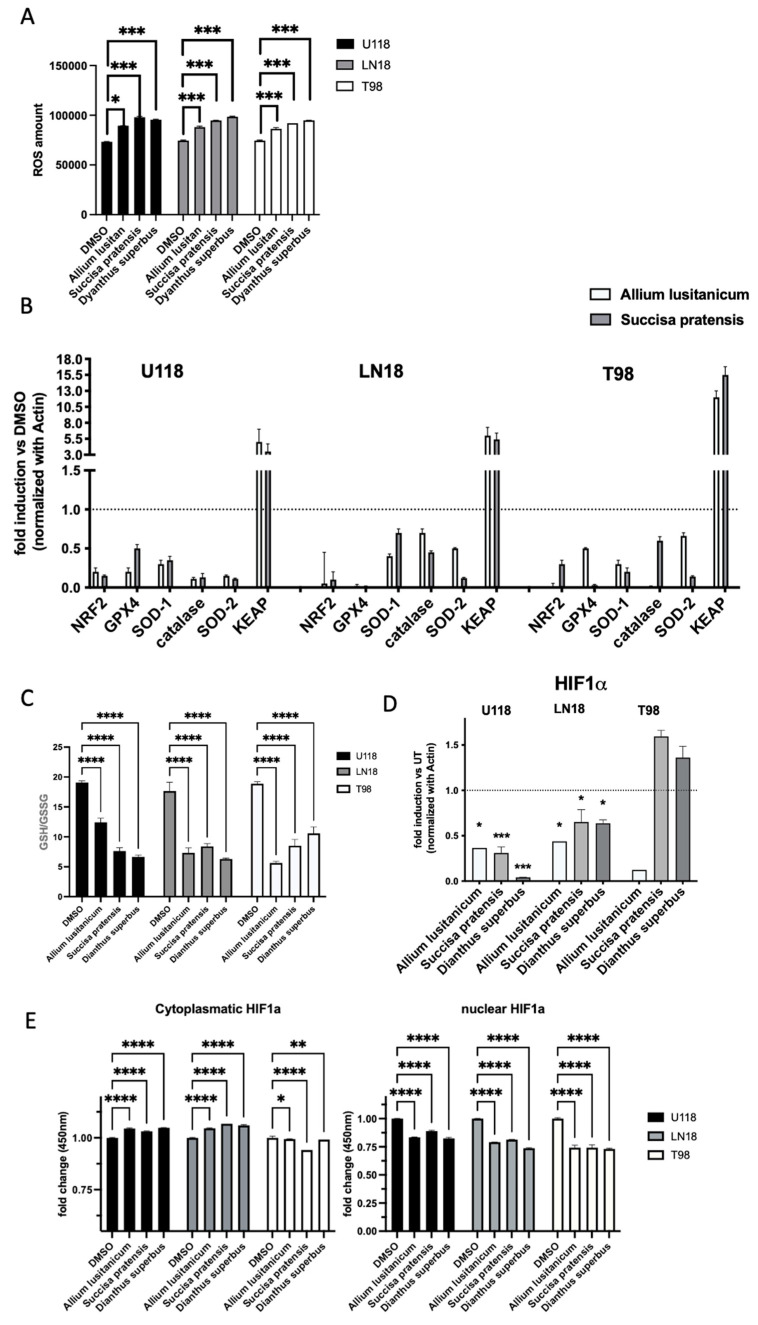
Plant extracts promoted oxidative stress and modulated target gene expression. (**A**) Representative graph of ROS content measured by ROS-Glo H_2_O_2_ Assay kit on resistant glioblastoma cell lines. (**B**,**D**,**F**,**G**) Real-time PCR was performed on a panel of genes involved in oxidative stress and inflammasome genes. Data were normalized to b actin, and the DDct values were expressed as fold of induction of the ratio between treated and not-treated cells. Data represent the mean ± SD of at least three independent experiments. * *p* < 0.05; ** *p* < 0.01; *** *p* < 0.001; **** *p* < 0.0001. ns, not significant (one-way ANOVA). (**C**) A representative graph of the GSH/GSSG ratio investigated by LC-MS analysis. (**E**) HIF-1α protein nuclear and cytoplasmatic extract quantification performed by ELISA-based kit The data are expressed as the amount of (O.D. 450 nm). Data represent the mean ± SD of at least three independent experiments. * *p* < 0.05; ** *p* < 0.01; **** *p* < 0.0001. (one-way ANOVA).

**Figure 5 nutrients-16-02389-f005:**
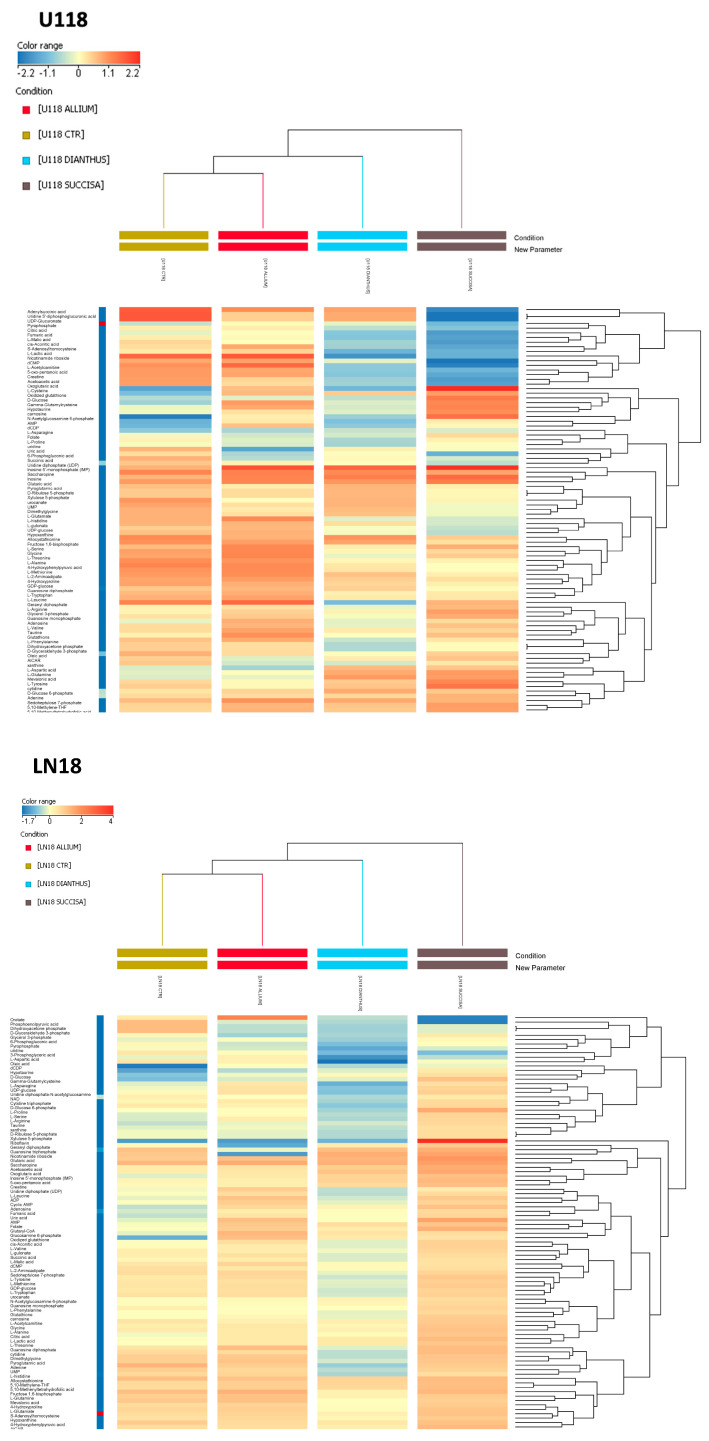
Biomolecules induced metabolic rewiring in GBM cell lines. Hierarchical clustering heatmaps of significantly different intracellular metabolites in U118 (upper panel), LN18 (middle panel), and T98 (lower panel), as detected by LC-MS. The lists of the different metabolites are derived from one-way ANOVA analyses. Colors represent different levels that increase from blue to red.

## Data Availability

The research group makes the data available to reviewers or other authors who make a request to the corresponding author for the data or access to the original analysis files.

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
