# Peer review of "A Novel Strategy for Glioblastoma Treatment by Natural Bioactive Molecules Showed a Highly Effective Anti-Cancer Potential"

_nutrients, 2024, doi:10.3390/nu16152389_

Round 1

Reviewer 1 Report

Comments and Suggestions for Authors

In the manuscript submitted to me for review entitled "A novel strategy for glioblastoma treatment by natural bioactive molecules showed a highly effective anti-cancer potential the authors Alessandro Giammona, Mauro Commisso, Marcella Bonanomi, Sofia Remedia, Linda Avesani, Danilo Porro, Daniela Gaglio, Gloria Bertoli  and Alessia Lo Dico present a study in which they used natural extracts of Allium lusitanicum, Succisa pratensis and Dianthus superbus as an alternative for the treatment and containment of glioblastoma (GBM).

They used a variety of methodologies, which are described in great detail, examining the problem from different points of view. The obtained results are presented using 5 figures in the main text of the manuscript and several additional figures and tables in the supplementary file.

The presented study may contribute to directing attention to the discovery of new natural products with antitumor activity affecting one of the most invasive tumors with the worst prognosis - GBM.

To support their research, the authors used 29 references that presented information from studies published mostly in the past two decades. About 2/3 of the total references are from the last 5 years, with a few from the current year 2024. This shows that the topic has been actively developed by many teams in recent years and would be of interest to Nutrients readers. I did not notice any redundant self-citations, all the references used are appropriate and necessary for the preparation of the manuscript.

My remarks and recommendations to the authors are:

1. Why did the authors choose to work with methanol extracts of the indicated plant species rather than ethanol, given the higher toxicity of methanol compared to ethanol?

2. In figure 2, the inscriptions are very small, and in 2C they are even of poor quality. If it is possible to increase the font of the inscriptions (if it is necessary to divide figure 2 into separate figures) and to improve the quality of 2C.

3. In figure 5, the inscriptions even at the highest magnification are unreadable and of poor quality. If it is possible to improve their quality.

4. In section 2.2. the authors claimed that the experiments performed did not show significant cytotoxicity against the cell lines studied, but later claimed that all cell lines treated with the plant extracts showed a change in cell morphology. How do the authors explain the two contradictory results? It is also not stated at what concentrations of the extracts these effects were observed. It is good to indicate in the text.

5. It is not stated by what methodology it was determined that the extracts did not show significant toxicity. It will be useful for readers if the methodology is indicated in the Materials and Methods section.

Reviewer 2 Report

Comments and Suggestions for Authors

The authors perform a biodiversity analysis to examine the potential efficacy of Allium lusitanicum, Succisa pratensis, and Dianthus superbus for improving glioblastoma.  Chromatograms are presented.  Results illustrate that the examined biomolecules promoted an antiproliferative and anti-migratory effect on cultured GBM cells. 

Experimental methods appear good and are well described.

Data analysis appears to be correct. However, the authors need to better clarify that log2 transformation was performed first followed by normalization given the order of operations is important to downstream results.  Also, the authors should better justify their choice for Pareto scaling. Pareto scaling is a dispersion-based method. However, it is very sensitive to large fold changes. This context needs to be explained and justified.

The paper would benefit with an "overview" figure that captures the general flow of experiments and the data analysis pipeline.

MINOR:

There is an excessive use of first person language for a scientific journal article in this domain.  Try to eliminate "we", "our", to only the places where there is no other alternative - such as brand new method development, etc.

The discussion would greatly benefit with the use of sub-headings to better organize topics.  Also, the authors should look at the structure of the paragraphs as some are 1/3 to nearly 1/2 a page while others are like 1 sentence. Use of sub-headings, bullets, or other formatting would increase readability and help to identify key themes important to the reader, like NFkB, ROS, inflammasones, hypotheses on how biomolecules are limiting cell cycle of glioblastoma, study limitations, future directions, etc.

Reviewer 3 Report

Comments and Suggestions for Authors

This article investigated the anti-cancer effect of Allium lusitanicum, Succisa pratensis, and Dianthus superbus using U118, LN18 and T98 cell lines. The content of the article is detailed, but I still have the following questions.
1. I suggest to clarify the research content and significance of this article at the end of the introduction.
2. Please check if the sentence 182-183 describes correctly? To my knowledge, the down-regulation of E-cadherin expression is a hallmark of EMT occurrence.
3. Why the inflammatory factors are reduced after Allium lusitanicum, Succisa pratensis, and Dianthus superbus? The relationship between inflammation and cancer requires a detailed introduction.
4. There are too many paragraphs in this article, and some parts do not need to be segmented.
5. In line 267-268, this doesn't seem to need to be divided into two parts.
6. In figure 2C, The text is too small to read clearly.
7. In figure 3, the scale bars is missing.
8. In figure 5, the text in hierarchical clustering heatmaps is too small to read clearly.
